# Proteomic screening of TMEM43 binding partners identifies VDAC leading to mitochondrial dysfunction

Qingqing Zhu[1◉], Guoxing Zheng◉[1◉*], Yingsi Lu◉[1◉], Yizhou Jiang[2], Yan Li[1], Hong Chen[1], Lifen Huang[1], Nannan Tang[1], Bo Li◉[3], Yi Lu[4*], Jian Zhang[4*], Chengming Zhu[1*]

1 Scientific Research Center, The Seventh Affiliated Hospital, Sun Yat-sen University, Shenzhen, Guangdong, China, 2 Institute for Advanced Study, Shenzhen University, Shenzhen, Guangdong, China, 3 Guangdong Provincial Key Laboratory of Digestive Cancer Research, Guangdong-Hong Kong-Macau University Joint Laboratory of Digestive Cancer Research, Scientific Research Center, The Seventh Affiliated Hospital, Sun Yat-sen University, Guangdong, Guangdong, China, 4 School of Medicine, Southern University of Science and Technology, Shenzhen, Guangdong, China

◉ These authors contributed equally to this work.

* zhuchm3@mail.sysu.edu.cn (CZ); zhangjian@sustech.edu.cn (JZ); opqsky@126.com (GZ); luy3@sustech.edu.cn (LY)

## Abstract

The transmembrane protein 43 (TMEM43) S358L mutation is the most common mutation found in patients with ARVC5, a severe subtype of arrhythmogenic right ventricular cardiomyopathy (ARVC). Characterizing the interacting proteins of TMEM43 mutants is important for unraveling the underlying mechanisms of the relative pathogenesis. Here, we applied quantitative immunoprecipitation (IP)-mass spectrometry (MS) to screen the differential binding partners between TMEM43 and TMEM43 p.S358L and identified 166 interacting protein candidates. Functional enrichment of these proteins highlighted their involvements of calcium signaling, lipid metabolism and cardiovascular diseases. Immunofluorescence (IF) staining and TurboID proximity-labeling further confirmed the interaction of several selected proteins. Moreover, we found that the interaction of both voltage-dependent anion-selective channel proteins (VDAC1 and VDAC2) to TMEM43 mutant significantly decreased. Importantly, lower VDAC binding mediated the mitochondrial dysfunction in cardiac myoblast H9c2 cells of TMEM43 p.S358L. The study provides a comprehensive view of the binding protein landscape of TMEM43 p.S358L, and implies a critical role of TMEM43 in mitochondria.

## Introduction

Arrhythmogenic right ventricular cardiomyopathy (ARVC) is a severe heart muscle disease leading to sudden cardiac death (SCD) [1]. ARVC is characterized with cardiac fibrofatty progress, ventricular arrhythmias, and ultimately heart failure [1].

**Data availability statement:** The mass spectrometry proteomics data have been deposited to the ProteomeXchange Consortium (http://proteomecentral.proteomexchange.org) via the iProX partner repository with the dataset identifier: PXD025904. All other relevant data are within the paper and its Supporting Information files.

**Funding:** This work was supported by grants from Shenzhen Science and Technology Planning Program (JCYJ20220530144817040 to C. Zhu, JCYJ20210324122814040 to G. Zheng, JCYJ20190809161811237 to J. Zhang and JCYJ20210324104214040 to Yi Lu), National Natural Science Foundation of China (81972420 to Yi Lu and 81972766, 82173336 to J. Zhang) and Chengming Zhu's start-up funding from Sun Yat-Sen University.

**Competing interests:** The authors have declared that no competing interests exist.

**Abbreviations:** T43, TMEM43; Mu, mutant; Vc, vector; ARVC, arrhythmogenic right ventricular cardiomyopathy; SCD, sudden cardiac death; IP, immunoprecipitation; IF, immunofluorescence; WT, wild type; DBPs, differential binding proteins; PCA, principal component analysis; MOI, multiplicity of infection; FDR, false discovery rate; GO, gene ontology; KEGG, Kyoto encyclopedia of genes and genomes; COG, clusters of orthologous groups of proteins; DSP, desmoplakin; JUP, junction plakoglobin; CAV1, caveolin 1; APOB, apolipoprotein B; ACTG1, γActin; CX43, connexin 43; EMD, emerin; SERCA2, sarcoplasmic/endoplasmic reticulum calcium ATPase 2; LMNA, lamin A/C; TMEM43, transmembrane protein 43; MS, mass spectrometry; IP3R, the inositol 1,4,5-trisphosphate (IP3) receptor; RYR2, ryanodine receptor 2; SR, sarcoplasmic reticulum.

Intense exercises and male gender are two risk factors for ARVC onset [2]. ARVC causes up to 25% of SCD in children, 22% of SCD in athletes and 11% of SCD in adults [3]. More than half of ARVC patients have the mutations of one of seven genes of the cardiac desmosome proteins [4], thus ARVC is defined as a 'disease of the desmosome' [5]. Desmosome proteins mediated the intercellular interaction between cardiomyocytes [5] and are required for normal cardiac muscle contraction [4,5]. In addition, several non-desmosomal genes including Tmem43, Desmin (DES) [6,7], Integrin linked kinase (ILK) [8], Filamin C (FLNC) [9], LEM domain containing protein 2 (LEMD2) [10] have been reported to be associated with ARVC.

TMEM43 is highly conserved and ubiquitously expressed in all organisms [11]. The Ser358Leu mutation of TMEM43 was firstly identified in 15 Newfoundland families of ARVC [12]. To date, the same mutation was found worldwide [13], and is referred as a common mutation of ARVC [14]. TMEM43 p.S358L mutation is associated with ARVC5, one severe subtype of ARVC with full penetrance, high percentage of death in patients [12]. Previous studies from ours and others proved that TMEM43 p.S358L mutant promoted cardiac adipo-fibrosis [15] and altered the localization of cardiac proteins [16]. It has been reported that the interaction proteins of TMEM43 play important roles in ARVC5 pathology [17]. Whereas the interaction protein networks of TMEM43 and how S358L mutation influence these protein interactions in ARVC remain to be explored.

Here, we systematically analyzed the binding partners of TMEM43 and TMEM43 S358L mutant, and identified the differential interaction proteins via quantitative immunoprecipitation (IP)-mass spectrometry (MS). Through the functional enrichment analyses, these differential binding proteins were found to be enriched on the pathways of calcium signaling, lipid transport and metabolism, and Wnt signaling that plays an important role in ARVC pathology. Immunofluorescence (IF) staining and TurboID biotin labeling experiments both validated the protein interactions found in IP-MS. Importantly, VDAC mediated mitochondrial dysfunction was found in cardiac myoblast cells of TMEM43 mutant. The findings expanded the understanding of the pathogenesis of ARVC through elucidating the binding protein networks of TMEM43 and TMEM43 S358L mutant.

## Materials and methods

### Cell lines

A549 cells obtained from America Tissue Type Collection (Manassas, VA; catalog # CCL-185) and cardiac myoblast H9c2 cells obtained from America Tissue Type Collection (Manassas, VA; catalog # CRL-1446) were both cultured with Dulbecco's Modified Eagle Medium (DMEM) containing 10% Fetal Bovine Serum (FBS), at 37°C, 5% CO2. The cells were infected with lentivirus carrying the p2k7 vector control, hemagglutinin (HA)-Flag-TMEM43 or HA-Flag-TMEM43 S358L mutant at MOI (multiplicity of infection) of 10 for 8h, and treated with blasticidin for 7 days to select the positively infected clones.

## Mice

Three to five adult wild type C57BL/6 mice were used in this study. All animals freely received food and water, housed in a temperature and humanity controlled environment with12-hour light/dark cycle. To reduce both the suffering and number of the animals, the principles of the 3Rs (replacement, reduction, and refinement) were followed. The health and behavior of the mice was monitored by a daily inspection. Endpoint criteria in all mouse studies were ≥ 20% weight lost of mice. Once mice reached the endpoint criteria, they were submitted to euthanasia immediately. No mice died before meeting criteria for euthanasia. The euthanasia of adult mice was performed as following: gradually introduce of pure CO2 at a rate of ~20% chamber volume per minute into the chamber for more than 5 minutes. Before removing the mice from the chamber, the death of mice was verified. All the mouse experiments were performed in accordance with the institutional guidelines and regulations, and approved by the Institutional Animal Care and Use Committees at Sun Yat-sen University (#SYSU-IACUC-2021-B1118). The study is reported in accordance with ARRIVE guidelines (https://arriveguidelines.org).

## Lentivirus packaging

On day 1, $4\times10^6$ 293T cells were seeded in a 10 cm dish. On day 2, the 293T cells reached 60%~80% confluence in the dish. A transfection mix was prepared as the following: 500uL 2x HBSS (Hank's Balanced Salt Solution) consisting of 3 plasmids: shuttle plasmid pMD2.G (1.25 µg), packaging plasmid psPAX2 (3.75 µg), expression plasmid DNA (5 µg). The mix was sub-sequentially added dropwise with 125µl of 2mM CaCl2 while vortexing, then incubate 30 min at room temperature. The mix was then added into the dishes, and the cells were maintained in an incubator at 5% CO2 and 37°C. The lentiviruses were harvested from the supernatant 48h after transfection.

## Quantitative Immunoprecipitation Mass Spectrometry (IP-MS) analysis

Immunoprecipitation: A549 cells transfected with vector control (triple), HA-Flag-TMEM43 wild type (WT, triple) and HA-Flag-TMEM43 S358L mutant (triple) were lysed with buffer containing 150mM NaCl, 50mM Tris-HCl (pH 7.4), 1mM Ethylenediaminetetraacetic acid (EDTA), 1% Triton X-100 and proteinase inhibitor cocktail (Roche, Ref#04693132001) on ice for 30 min, then centrifuged at 12500 rpm at 4°C for 10 min. The protein concentration in the supernatants was determined by PierceTM BCA protein assay kit (thermal scientific, Ref#23227). The cell lysates of same protein amounts were immuno-precipitated (IP) with ANTI-FLAG@ M2 Affinity Gel (Sigma, Cat# A2220) and eluted by 3XFlag peptides (Sigma, Cat# F3290) at concentration of 150ng/ul. The elution was immunoprecipitated again with EZviewTM Red Anti-HA Affinity Gel (Sigma, Cat# E6779). Immunoblotting of HA (cell signaling technology, Cat#3724, 1:1000 4°C blot overnight) to the products of each step was performed to ensure the working of IP procession.

Trypsin digestion: Proteins were disulfide reduced with 30mM dithiothreitol (DTT) and alkylated with 55mM iodoacetamide. The protein concentration was determined by Bradford method. 100ug protein of each sample was digested by trypsin (trypsin: protein = 1:100) at 37°C for more than 8 h. The obtained digestion products were acidified with an equal volume of 0.1% formic acid (FA) and mixed with 10 mg methanol activated C18 column material (Phenomenex, Cat#8B-S100-KEG). The sample was then washed twice with 0.1% FA + 3% Acetonitrile (ACN) for desalting and eluted with 1 mL 0.1% FA + 80% ACN. The eluted peptide was dried with a vacuum concentrator.

Liquid Chromatography (LC)-MS/MS: peptide samples were diluted to 1ug/ul by 5ul on-board buffer. The peptides were scanned with a mass-to-charge ratio of 350–1500 by the scanning mode of 120 min. With mobile phase A solution (98% water, 2% ACN, 0.1% FA), B solution (98% ACN, 2% water, 0.1% FA), pre-column (300um × 0.5 mm, 3 um), analytical column (3 um, 75um × 150 mm, Welch Materials, Inc), and spray voltage 1.9KV, peptides separated by liquid phase were ionized by nanoESI source and entered into tandem mass spectrometer Q-Exactive HF-X (Thermo Fisher Scientific, San Jose, CA) detection to generate raw data, which had the following main parameter settings: ion transfer tube temperature

320°C, scanning range 350-1500m/z, primary resolution 60000, C-Trap 3e6, IT 80ms; secondary resolution 15000, C-Trap1e5, IT 100ms, CE28; threshold intensity 1.0e4, dynamic exclusion 30s.

Data Processing: Maxquant (version 1.6.17.0) was utilized to search MS/MS spectra from each raw data file against the UniProt human database (20366 entries) and the msconvert module in Trans-Proteomic Pipeline was used to convert raw files into mgf files. The search criteria consisted of full tryptic specificity, allowing two missed cleav-ages, an initial mass tolerance of 20 ppm. Carbamidomethylating on cysteine was set as fixed modification, oxidation on methionine and Acetyl on the N-terminal of protein was set as variable modifications. The "hit" peptides were chosen via a high confidence score filter False Discovery Rate (FDR) < 1%. For normalizing the different protein abundances in different experiments, label free quantitative was used by Maxquant to calculate the final median-normalized protein abundance. To quantity compare the differential binding proteins of TMEM43 versus its S358L mutant, T-test of the triple samples in both groups was determined, in which p value < 0.05 is considered significant. For detection of TMEM43 binding partners, the same test was used and those proteins with Vc area zero but T43 area not zero were also included.

The mass spectrometry proteomics data have been deposited to the ProteomeXchange Consortium [18] (http://proteomecentral.proteomexchange.org) via the iProX partner repository [62] with the dataset identifier: PXD025904.

## Enrichment analysis of differential binding proteins

We generated protein sets from the IP-MS data by setting the cut-off fold change larger than 2.0. Gene Ontology (GO) enrichment analysis was conducted using the KOBAS 3.0 web server to identify biological processes that were enriched within differential binding proteins. GO terms with corrected P value less than 0.05 were considered significantly enriched. Additionally, KOBAS 3.0 tool was applied for other types of enrichment analysis. Kyoto Encyclopedia of Genes and Genomes (KEGG) PATHWAY, Reactome, BioCyc and PANTHER database were used in the metabolic pathways enrichment analysis. Significantly enriched diseases were also identified by the tool using Online Mendelian Inheritance in Man (OMIM), KEGG DISEASE and National Human Genome Research Institute (NHGRI) Genome-Wide Association Studies (GWAS) Cata-log database, with a corrected P value less than 0.05.

## Immunofluorescence (IF) staining

A549 or H9c2 cells stably expressing empty vector, TMEM43 or its S358L mutant were cultured overnight ($2 \times 10^5$ cells/well) on coverslips in 24-well plate, rinsed with PBS, and fixed in 4% paraformaldehyde for 10 min. The cells were permeabilized with 0.25% Triton X-100 in PBS for 15 min. Cells were then blocked for 1 hour at room temperature in 5% BSA in PBS and incubated overnight at 4°C in 2.5% BSA in PBS with indicated antibodies including anti-Flag (Abmart; 1:500), anti-HA (Abmart; 1:500), an-ti-ACTG1 (Novus Biologicals), anti-CX43 (Cell Signaling Technology), anti-EMD (Cell Signaling Technology), anti-Caveolin1 (Cell Signaling Technology), anti-APOB (Bioss), anti-LUMA (Cell Signaling Technology) and anti-ATP2A2 (Abcam). After washing, slides were incubated for 1 hour at room temperature with anti-mouse Alexa-488 (Invitrogen; 1:600) or anti-rabbit Alexa-546 (Invitrogen; 1:600). After washing, cells were counterstained with 1ug/mL DAPI (Beyotime; 1:5000) for nuclei. The hearts of adult wild type of C57BL/6 mice were collected, frozen in Optimal Cutting Temperature (OCT) and sectioned at 10um thickness. The sections were fluorescently stained with antibodies of TMEM43 (Santa Cruz), VDAC1 (ABclonal) and VDAC2 (Proteintech). Cells were imaged on a Zeiss LSM780 laser scanning confocal system (Carl Zeiss MicroImaging, LLC.) or Leica DM6B Fluorescent Microscope (Leica Microsystems, Germany) with 3 channels of Alexa-488 (green, excitation wave 488nm, emission wave 551nm), Alexa-546 (red, excitation wave 543nm, emission wave 659nm) and DAPI (blue, excitation wave 405nm, emission wave 459nm). For the Pearson's colocalization correlation coefficient (PCC) analysis, the overlap of the two channels (Alexa-488 and Alexa-546) was assessed across the whole image using the Colocalization Tools in ZEN Blue (Carl Zeiss MicroImaging, LLC.). PCC values of quadrant #3, containing relatively high intensities of both channels, were detected. Values can range from −1 (an "anti-colocalization" relationship), to 0 (a random cloud of no relationship), or +1 (a perfect colocalization).

## TurboID Plasmids

The homology regions were generated by PCR on pCDNA3-TMEM43/TMEM43-S358L plasmids in our lab and then separately subcloned into the pCDNA3-3xHA-TurboID vector (Addgene, #107171) with a homology cloning kit (Vazyme, #C112). The restriction sites of EcoRI and XbaI were used for cloning.

## TurboID mediated proximity labeling [19]

For immunofluorescent staining, $6 \times 10^4$ H9c2 cells were seeded on glass coverslips in a 24-well plate for transfection next day. 1 μg fusion constructs were introduced into cells by transfection with liposomal reagent (Yeason, #40802ES02). 36h post-transfection, the culture medium was replaced by complete DMEM media supplemented with 50μM biotin (MCE) for labeling of 30 min. Then the labeling media was removed, and cells were carefully washed for five times with 500 μL cold PBS. After fixation with paraformaldehyde and permeabilization with cold methanol, H9c2 cells were stained with the following primary antibodies: rabbit mAb anti-HA-tag (Cell Signaling Technology, #3724), rabbit mAb anti-SERCA2 ATPase (Abcam, #150435), mouse mAb an-ti-ApoB (Biorbyt, #orb69428), rabbit mAb anti-Emerin (Cell Signaling Technology, #30853), rabbit pAb anti-VDAC2 (Proteintech, #116663–1-AP), rabbit pAb an-ti-Caveolin-1 (Cell Signaling Technology, #3238), mouse mAb anti-Lamin A/C (Cell Signaling Technology, #4777), rabbit pAb anti-Connexin 43 (Cell Signaling Technology, #3512), rabbit pAb anti-VDAC1 (ABclonal, #A15735), and with the following secondary antibodies: Alexa Fluor 488 Streptavidin (Yeason, #35103ES60), goat anti-rabbit IgG (H + L) cross Alexa Fluor Plus 594 (Thermo Fisher, #A32740), goat anti-mouse IgG (H + L) cross Alexa Fluor Plus 546 (Thermo Fisher, #A-11030). 1 μg/ml DAPI was used for nuclei co-staining.

For pull-down of biotinylated proteins, $5 \times 10^6$ H9c2 cells were incubated with 50μM biotin for 30 min, washed by cold PBS and lysed by 500 μL RIPA lysis buffer at 36h after transfection. Lysates were sonicated and centrifuged at 12,000 rpm for 10 min, the supernatant was collected. 20 percent of the products was taken out as input sample. The protein concentrations of products were determined with BCA assay kit (Thermo Scientific, #23225). Suitable amounts of products were incubated with 200 μL of streptavidin magnetic beads (MCE, #HY-K0208) overnight at 4°C to enrich biotinylated proteins. Subsequently, the beads were washed twice with 1mL RIPA lysis buffer (2 min), once with 1 mL 0.1 M Na2CO3 solution (10 sec), once with 1mL 2M urea in PH 8.0 Tris-HCl (10 sec), and then twice with 1 mL RIPA lysis buffer (2 min). After washing, the beads were resuspended in 3 x protein loading buffer supplemented with 2mM biotin and 20mM DTT, then boiled at 100°C for 10 min. Finally, the eluate of the beads was collected by a magnetic rack and further analyzed with Western blotting to detect the biotinylated proteins.

## Immunoprecipitation (IP)

H9c2 cells transfected with vector control, HA-Flag-TMEM43 wild type and HA-Flag-TMEM43 S358L mutant were lysed on ice for 30 min, then centrifuged at 12500 rpm at 4°C for 10 min. The protein concentration in the supernatants was determined by PierceTM BCA protein assay kit (thermal scientific, Ref#23227). The cell lysates of same protein amounts were immunoprecipitated (IP) with ANTI-FLAG@ M2 Affinity Gel (Sigma, Cat# A2220) and wash by buffer for 3 times. The beads were supplemented with protein loading buffer and boiled for 5 min. Immunoblotting of VDAC1 (Abclonal, Ref# A15735), VDAC2 (proteintech, Cat#11663–1-AP) and Flag (Ab-mart; 1:500) to the products of input and IP was carried out. According to the molecular weight of target proteins, the membranes were cropped prior to hybridizations with antibodies. The original, unprocessed images of all blots with membrane edges visible but not the full length membranes are provided in the supplementary files.

Adult C57/B6 mice were sacrificed and the hearts were collected and immediately frozen in liquid nitrogen. The frozen hearts were then homogenized and lysed on ice with IP lysis buffer for 30 min. After centrifuge, the supernatants were collected and the protein concentration was detected. The cell lysates of same protein amounts were immunoprecipitated

(IP) with antibodies of TMEM43 (Santa Cruz, Cat# sc-365298), VDAC1 (Santa Cruz, Cat# sc-390996) and wash by buffer for 3 times. The beads were supplemented with protein loading buffer and boiled for 5 min. The products were submitted to immunoblotting.

### Mitochondria staining

H9c2 cells were incubated with 200nM MitoTracker Red CMXRos (C1035, Beyotime) at 37°C for 20 min, and washed with HBSS, then fixed with 4% paraformaldehyde for 15 min at 37°C. The pictures of mitochondrial network morphology were taken under a confocal microscope (Nikon, Japan) using a 100x oil immersion lens.

### Detection of mitochondrial membrane potential

H9c2 cells were incubated with JC-1 (C2005, Beyotime) at 37°C for 30 min, then washed with PBS. The signals of JC-1 aggregate (red, 525/590 nm) and JC-1 monomer (green, 485/530 nm) were subjected to detection under fluorescence microscope or flow cytometry analyses. The data from triple flow cytometry analyses and the ratio of JC-1 aggregate/ JC-1 monomer (indicate the mitochondrial membrane potential – Δψm) was calculated.

### Detection of Ca2+level

For $Ca^{2+}$ level assay, H9c2 cells were washed twice with PBS, and then stained with 10μM Fluo4AM fluorescent probe (Servicebio, G1724-100T) at 37°C for 30 min. After PBS washing, the intracellular $Ca^{2+}$ level was determined under a laser confocal microscope. The excitation wavelength and the emission wavelength were 488nm and 525nm, respectively.

### Measurement of mitochondrial ROS

For mitochondrial ROS level detection, H9c2 cells were washed with PBS and then incubated with 5 μM MitoSOX (Life Technologies, M36008) for 15 min at 37°C. After washed with PBS, cells were imaged with confocal microscopy.

### The statistical analyses

To quantity compare the differential binding proteins of TMEM43 versus its S358L mutant, T-test of the triple samples in both groups was determined in IP-MS analysis, in which P value < 0.05 is considered significant. For detection of TMEM43 binding partners, the same test was used. Those proteins found in T43 cells but not in vector cells were also indicated as T43 binding partners. In all the enrichment analyses, the terms with corrected P value less than 0.05 were considered significant.

### Ethics approval

The animal protocols were approved by the Animal Care and Experimentation Committee of Sun Yat-Sen University (#SYSU-IACUC-2021-B1118).

## Results

### Systematic analysis of the protein networks binding to TMEM43 and its S358L mutant via IP-MS

In order to elucidate the protein regulatory network in ARVC, we systematically investigated the binding protein network of TMEM43 and the S358L mutant. A549 cells transfected with empty vector control, HA-Flag-TMEM43 WT and HA-Flag-TMEM43 S358L mutant were lysed, immunoprecipitated with Flag antibody and eluted by Flag peptide. For maximum reduction of nonspecific binding of Flag antibody, the elute was immunoprecipitated again with HA antibody and then subjected to mass spectrometry (Fig 1A). The IP efficiency of the three cell lines was tested via immunoblotting with HA antibody (S1A Fig), which showed a specific band of molecular weight at about 45kD in both cell lines of TMEM43 and

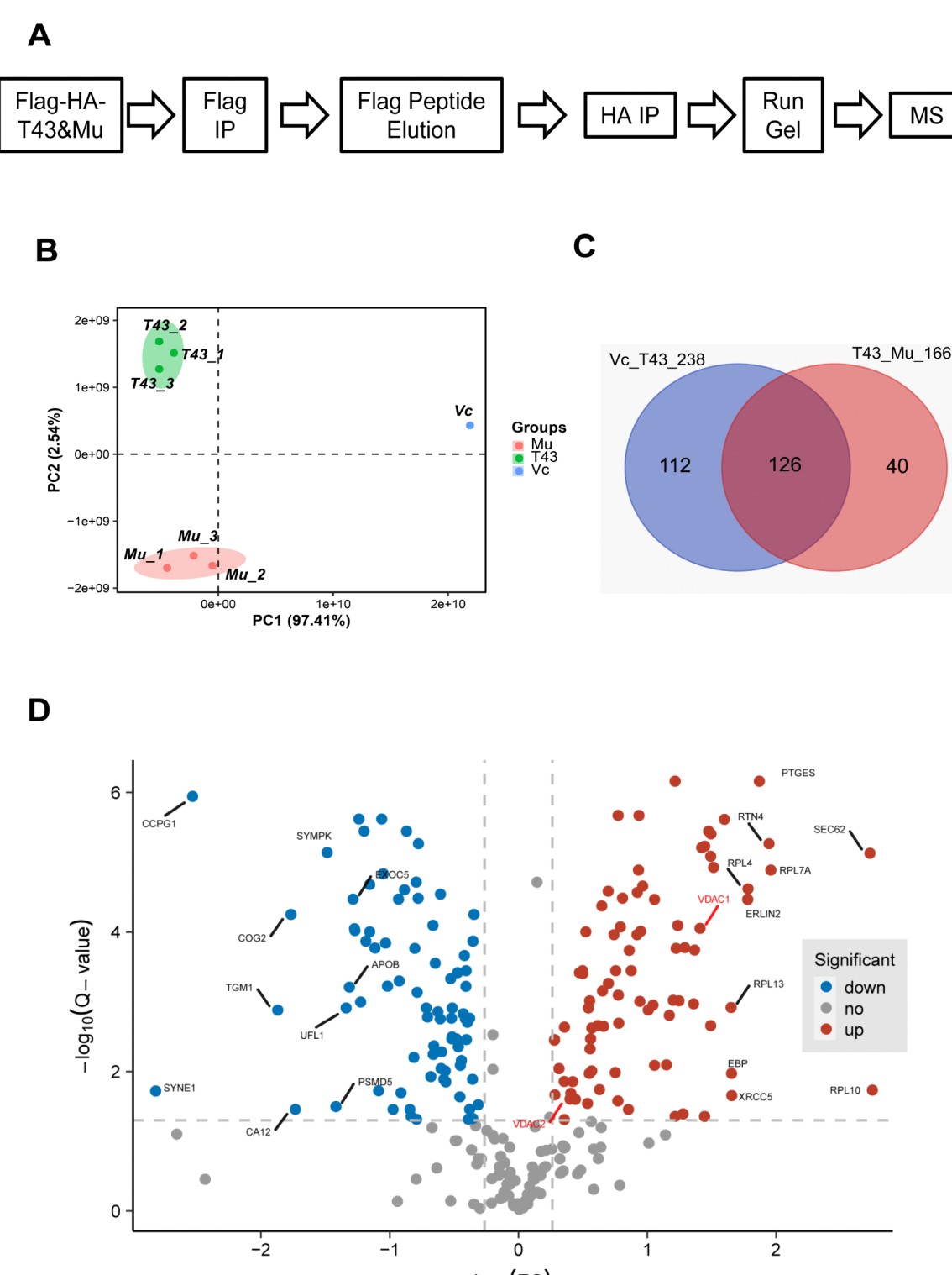

**Fig 1. IP-MS systematically determined the differential binding differential binding proteins of TMEM43 versus TMEM43 p.S358L mutant.**
(A) The flow chart of IP-MS assay. (B) Principal Component Analysis (PCA) of MS result showed the three clusters of the IP samples, suggesting the significant differentiation between the three groups. T43 TMEM43; Mu Mutant; Vc Vector (C) Venn diagram showed 112 and 40 proteins were found to

bind to TMEM43 and TMEM43 **p.**S358L mutant respectively, and another 126 proteins were seen to bind to both of them. **(D)** The volcano plot of DBPs in TMEM43 versus TMEM43 **p.**S358L mutant. Important features were determined by volcano plot with fold change (FC) threshold and t-tests threshold. The blue dots and red dots represent the proteins of significantly decreasing binding and increasing binding ability of TMEM43, respectively.

its S358L mutant but not in empty vector control cells. The target proteins in input lysates (input in S1A Fig) were eluted out (E1-3 in S1A Fig) from Flag beads and further enriched by HA antibody (HA in S1A Fig). Silver staining of the final IP products confirmed the target proteins in both cell lines of TMEM43 and its S358L mutant but not in empty vector cells (S1B Fig).

Principal Component Analysis (PCA) of MS results showed three distinct clusters of the IP samples (Fig 1B), suggesting the differentiation between the three groups. The proteins with p value < 0.05 in T-test of the triple samples in wild-type TMEM43 and its S358L mutant cells were referred as differential binding proteins (DBPs). Totally 238 proteins were found to interact with TMEM43 (S1 Table), while 166 DBPs between TMEM43 and its S358L mutant were identified (S2 Table). 126 proteins were common between the 2 groups as shown in the Venn diagram (Fig 1C). The volcano plot showed the significant differences of DBPs in TMEM43 versus TMEM43 p.S358L mutant (Fig 1D).

## Annotation of DBPs uncovered important ARVC related processes

KEGG pathway annotation of DBPs (S2A Fig) revealed some ARVC related pathways, including dilated cardiomyopathy, tight junction and calcium signaling pathway. Gene ontology (GO) enrichment of DBPs (S2B Fig) suggested the importance of Wnt signaling, fatty acid metabolism and heart contraction, all of which are involved in ARVC pathogenesis. These results suggest TMEM43 S358L mutation may be involved in ARVC pathology through the alteration of TMEM43 binding protein network.

To analyze the potential functions of the DBPs, the enrichment analyses of KEGG pathway and human diseases were conducted. ARVC, hypertrophic cardiomyopathy, dilated cardiomyopathy, tight junction, and calcium signaling pathway were found in KEGG pathway enrichment (Fig 2A). ARVC, muscular diseases, lipid/glycolipid metabolism, and cardiovascular diseases were determined in disease enrichment (Fig 2B). These results suggest that TMEM43 S358L mutant may influence ARVC development through the pathways including lipid metabolism, tight junction, and calcium signaling pathway.

## The immunofluorescence staining and the TurboID experiment verified the DBPs

In order to validate the DBPs, we performed immunofluorescence staining on A549 cells and H9c2 cardiac myoblasts cells stably expressing HA-Flag-TMEM43 and its S358L mutant. Several DBPs, including Caveolin 1 (CAV1, function in heart contraction), γ-actin (ACTG1, muscle fibers), Apolipoprotein B (APOB, lipid metabolism), that may be associated with ARVC were selected. The staining showed that all these proteins partially co-localized with HA-Flag-TMEM43 and its mutant on the envelopes of nucleus in A549 cells (Fig 3A, green HA, or red Flag), and statistics of 7 random regions indicated the different co-localization (yellow) percentage of these proteins in WT and mutant TMEM43 cells (Fig 3B). The Pearson's correlation coefficient (PCC) of green and red channels were > 0.95 (indicating the co-localization) in all TMEM43 and its mutant cells. Similar staining pattern of some of these proteins and TMEM43 were also found in H9c2 cardiac myoblasts (S3 Fig). Taken together, the different co-localization percentage of the selected binding partners confirmed their differential binding affinity to TMEM43 and its S358L mutant.

In order to further confirm the interaction of TMEM43 with its binding partners, we constructed 3xHA-Turbo -TMEM43 and its S358L mutant in pCDNA3 plasmids (S4A Fig), then transfected these plasmids into H9c2 cardiac myoblasts. The obtained cells were labelled with biotin and submitted to streptavidin immunoprecipitation (IP) or immunofluorescence staining with streptavidin and indicated antibodies. Immunoblotting of immunoprecipitated products showed that wild-type

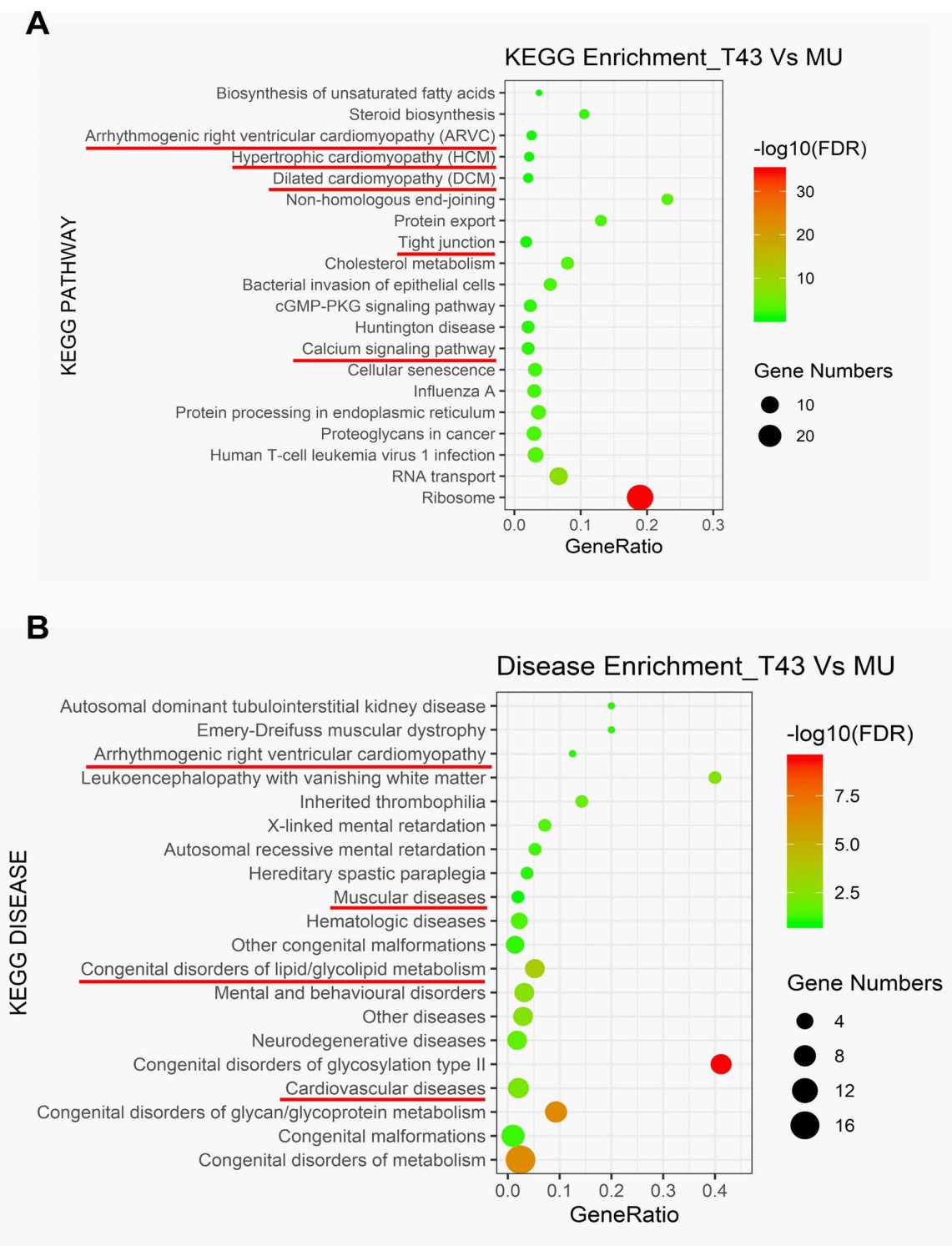

**Fig 2. The pathway enrichments of differential binding proteins of TMEM43 between cells expressing TMEM43 and TMEM43 p.S358L mutant.** (A) Scatterplot of enriched KEGG pathways of DBPs in TMEM43 versus TMEM43 **p.**S358L mutant. (B) Scatterplot of enriched diseases of DBPs in TMEM43 versus TMEM43 **p.**S358L mutant. Both enrichments highlight arrhythmogenic right ventricular cardiomyopathy (ARVC) and cardiovascular diseases. The permission of KEGG usage in this article was kindly provided by Kanehisa Laboratories [20].

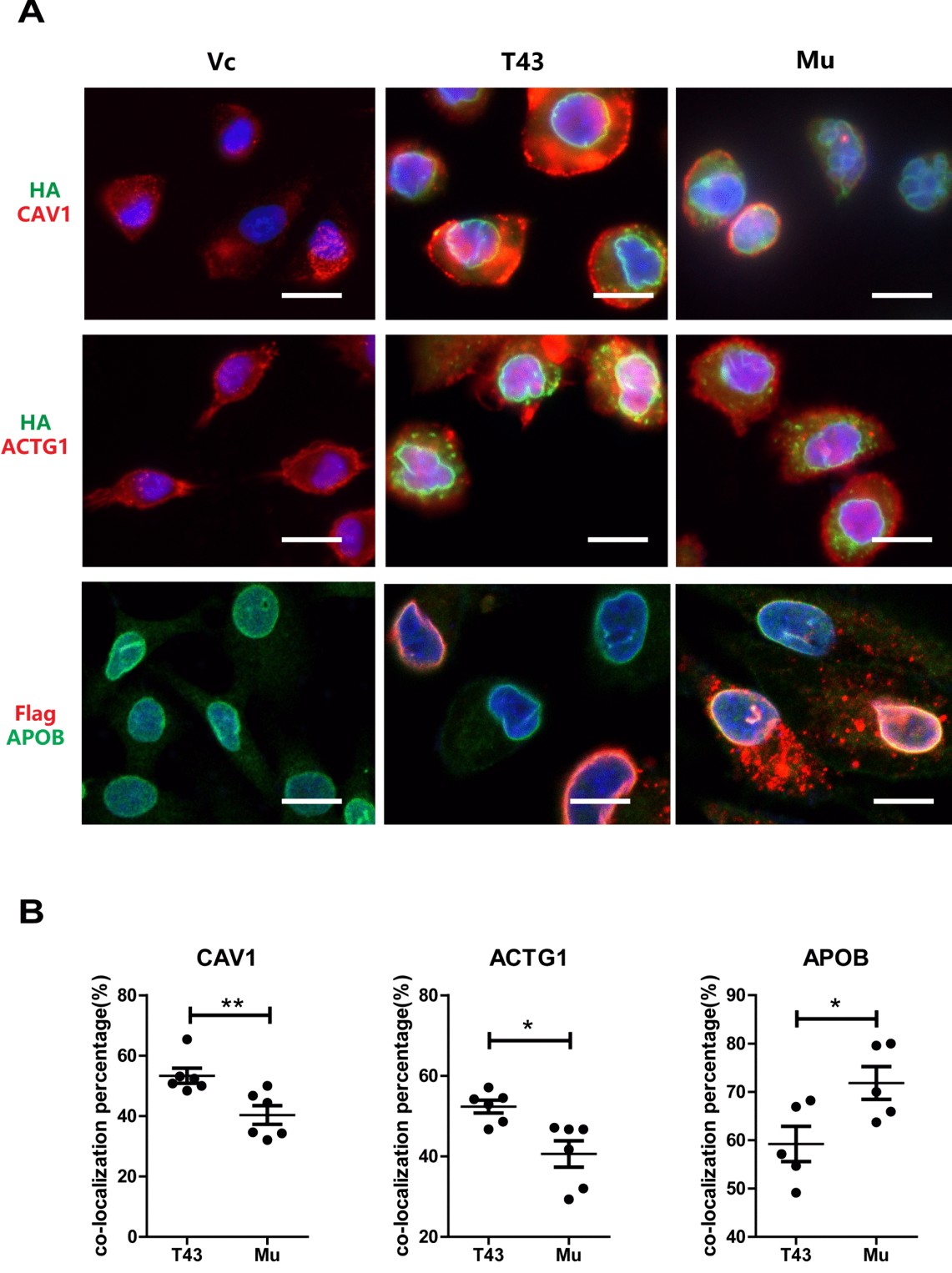

**Fig 3. The interaction of TMEM43 and its S358L mutant with several selected binding partners were confirmed in A549 cells. (A)** The Immunofluorescence (IF) staining showed the partially colocalization of HA-Flag-TMEM43 and its S358L mutant with the selected binding partners including γActin (ACTG1), Caveolin 1 (CAV1), and APOB in A549 cells. A549 cells stably expressing HA-Flag-TMEM43 and its S358L mutant were stained with DAPI, HA antibody or Flag antibody, and the indicated antibodies. The pictures were taken under fluorescent microscope. scale bar: 10um **(B)** The colocalization percentage in 7 random regions of IF staining under 200 magnifications was counted and compared. Student t-test was performed. * indicates p<0.05, ** indicates p<0.01.

TMEM43 and its mutant (indicated by HA) were successfully immunoprecipitated by streptavidin (S4A Fig), this validated the working of the TurboID system. While no interaction of indicated proteins was found in the empty transfected H9c2 control cells, the binding of proved TMEM43 partners Lamin A/C (LMNA), Emerin (EMD), and Sarcoplasmic/Endoplasmic Reticulum Calcium ATPase 2 (SERCA2 or ATP2A2) was detected in cells transfected with wild-type TMEM43 and its mutant (S4A Fig). IF staining of TurboID H9c2 cells revealed that the signals of biotin and HA labelled TMEM43 were completely overlapping (S5 Fig), indicating the precision of TurboID system. Furthermore, the nuclear membrane location biotin was partly overlapped with the indicated proteins including APOB (S4B Fig), ATP2A2, EMD, and LMNA (S5 Fig), suggesting the interaction of these proteins with TMEM43 and its mutant. Thus, the interaction of TMEM43 with its binding partners discovered in IP-MS was further validated in the TurboID system.

### Lower VDAC binding and the mitochondrial dysfunction in the TMEM43 p.S358L H9c2 cells

We noticed that both VDAC1 and VDAC2 were up-regulated in WT TMEM43 cells (Fig 1D), and they are key molecules in mitochondrial calcium ion transmembrane transportation. Furthermore, calcium signaling pathway (Fig 2A, and S2A Fig) showed up several times in functional enrichment analyses. Thus, we asked whether TMEM43 mutant influence on mitochondria and calcium signaling. We firstly verified the differential interaction of VDACs with TMEM43 and its mutants. The immunofluorescence staining of HA-TMEM43 and VDACs displayed relative less co-localization ratio in the mutant cells at the bottom (Fig 4A and 4B). Moreover, Flag immunoprecipitation (IP) exhibited the lower binding of both VDAC1 and VDAC2 in mutant H9c2 cells (Fig 4C). Furthermore, the physiological interaction of TMEM43 with VDAC1 were validated by endogenous IP of mouse hearts (Fig 4D). Together, these data suggest that the interaction of TMEM43 p.S358L with VDACs declined.

To dissect the function of TMEM43 in the mitochondria, the specific mitochondrial dye – MitoTracker red staining were performed. TMEM43 p.S358L caused the mitochondrial fragmentation indicated by a smaller number of mitochondrial branches (Fig 5A). To further evaluate mitochondrial function, we performed JC-1 staining to determine $\Delta \psi m$. Both imaging (S10 Fig) and flow cytometry (Fig 5B) exhibited that the JC-1 aggregate/monomer ratio decreased, which indicated $\Delta \psi m$ loss in mutation cells. Moreover, the mitochondrial matrix $Ca^{2+}$ level decreased and the ROS level significantly increased in mutant cells (Fig 5C), suggesting the mitochondrial dysfunction. Taken together, TMEM43 p.S358L caused the mitochondrial dysfunction in H9c2 cells.

## Discussion

ARVC is a severe cardiac muscle disease, featured with cardiac fibrofatty, arrhythmias, and always leading to sudden cardiac death. Sequencing of the genome of ARVC5 patients revealed genetic mutation of TMEM43 S358L. However, how the protein networks mediate the effect of TMEM43 p.S358L in ARVC remains to be explored. In this study, we applied quantitative IP-MS to distinguish the variation of interacting protein networks between TMEM43 and TMEM43 p.S358L. Systematic analysis of the protein networks binding to wild-type TMEM43 and its S358L mutant via IP-MS identified 238 specific proteins interacting with wild-type TMEM43 and 166 DBPs respectively.. Functional enrichment analysis revealed that the DBPs are involved in various facets of ARVC pathogenesis, including tight junction, Wnt signaling, lipid metabolism, calcium signaling, and arrhythmias relative cardiac muscle contraction. The results emphasized the central roles of TMEM43 p.S358L in ARVC.

VDAC1 and VDAC2 are associated with mitochondrial calcium ion transmembrane transportation [21]. The interaction between VADC1 and The inositol 1,4,5-trisphosphate (IP3) receptor (IP3R) enhanced the transportation of $Ca^{2+}$ from sarcoplasmic reticulum (SR) – the major intracellular calcium storage organelle, to mitochondria and induces apoptosis of cardiomyocytes [22]. Prevention of VDAC1 upregulation in ischemia reperfusion (I/R) resulted in a smaller infarct size and was beneficial for the inhibition of heart failure progress, through a blockage of mitochondrial $Ca^{2+}$ uptake [23]. Physiologically, the VDAC2-ryanodine receptor 2 (RYR2) couple [24] also promoted the shuttling of $Ca^{2+}$ from SR to mitochondria,

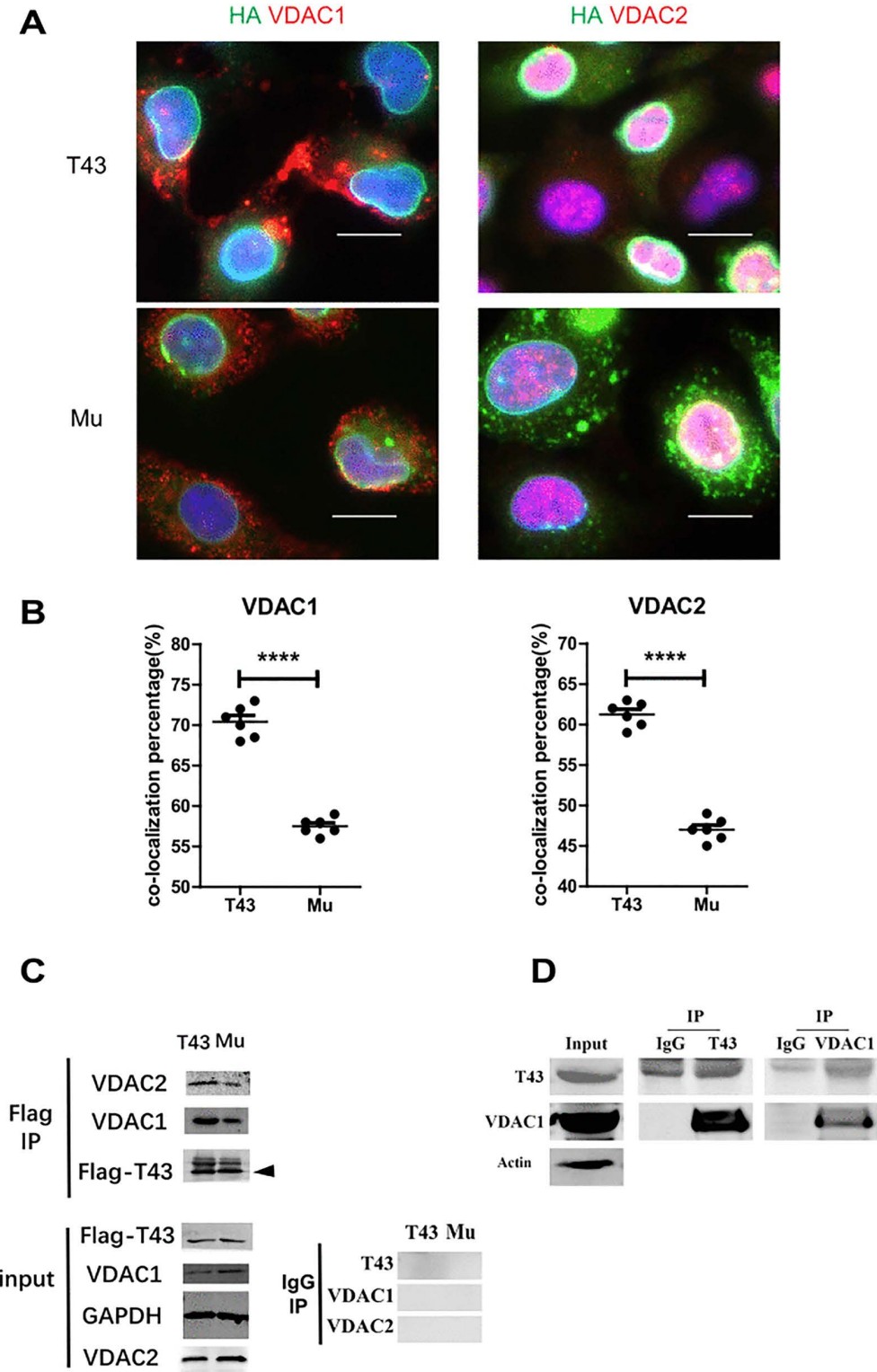

**Fig 4. S358L mutation reduced the interaction of TMEM43 with VDACs. (A)** IF staining showed the colocalization of Flag-TMEM43 and its S358L mutant (green) with VDAC1 and VDAC2 (red) in cardiac myoblast H9c2 cells. H9c2 cells stably expressing HA-Flag-TMEM43 and its S358L mutant were stained with DAPI, and the indicated antibodies. Scale bar: 10um **(B)** The colocalization percentage in 6 random regions of IF staining under 200 magnifications was counted and compared. **(C)** Immunoblotting with indicated antibodies displayed the protein expression of VDAC2, VDAC1 and Flag-TMEM43 before and after Flag IP. **(D)** The lysate of mouse hearts was IP with indicated antibodies and submitted to immunoblotting.

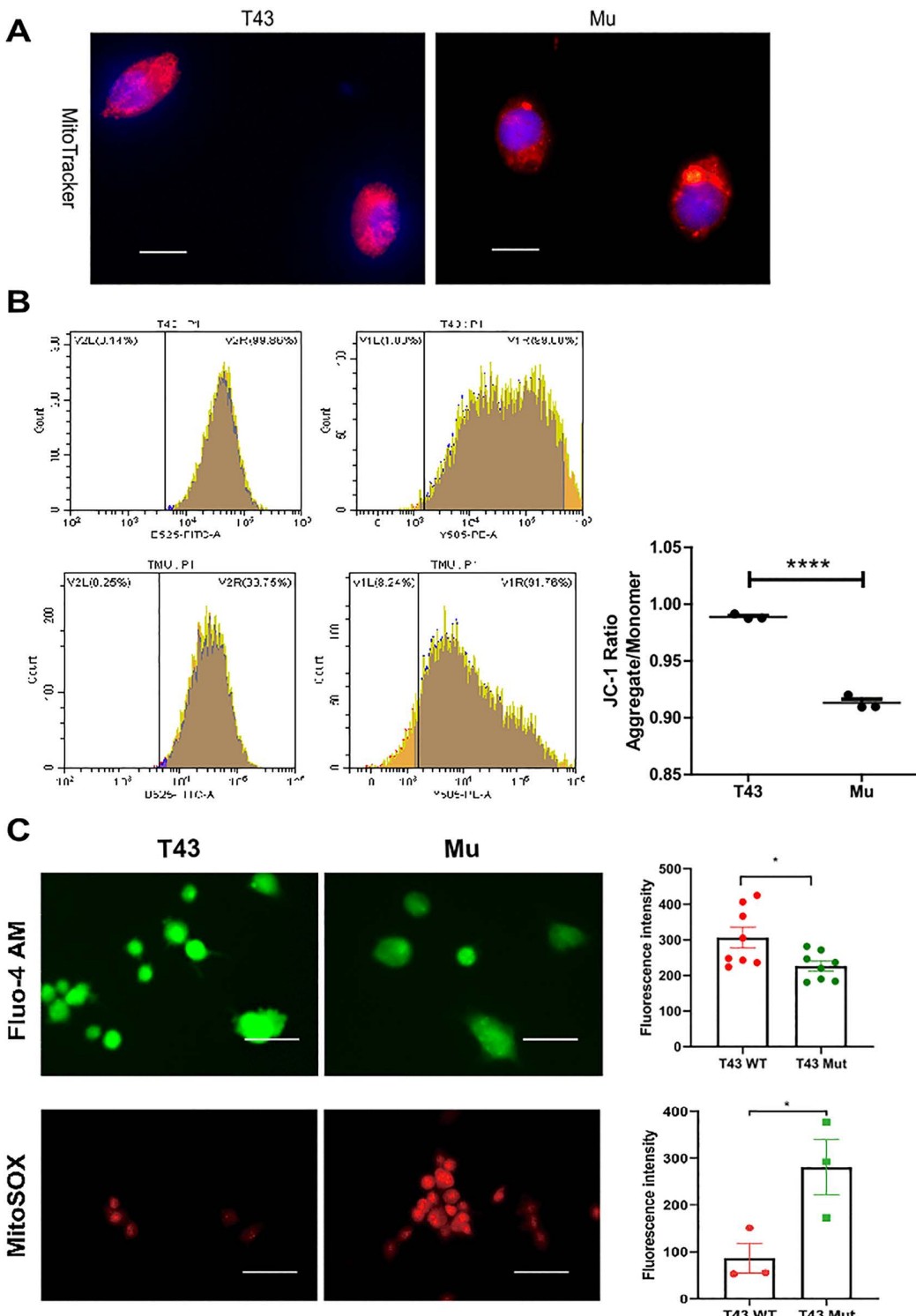

**Fig 5. Mitochondrial dysfunction in the TMEM43p.S358L H9c2 cells. (A)** MitoTracker red staining indicated the mitochondrial morphology in H9c2 cells stably expressing HA-Flag-TMEM43 and its S358L mutant. H9c2 cells stably expressing HA-Flag-TMEM43 and its S358L mutant were stained with DAPI, and MitoTracker red. Magnification 100X, Scale bar: 4um **(B)** Flow cytometry exhibited the cell percentage of JC-1 monomer (FITC) and aggregate (PE). Quantification of mitochondrial membrane potential (the ratio of JC-1 aggregate/ monomer) was performed with triple analyses of flow

cytometry. **(C)** The Ca2+ level and mitochondrial ROS level in TMEM43 and mutant H9c2 cells. Fluo-4 AM fluorescent staining (top panel) indicated the intracellular Ca2+ level in H9c2 cells stably expressing HA-Flag-TMEM43 and its S358L mutant. MitoSOX fluorescent staining (bottom panel) indicated the mitochondrial ROS level in H9c2 cells stably expressing HA-Flag-TMEM43 and its S358L mutant. Magnification 100X, Scale bar: 100um. Quantification of fluorescence intensity was performed for both assays.

but provided energy supply for cardiac contraction. Knockout of VDAC2 decreased mitochondrial Ca$^{2+}$ uptake and led to cytosolic Ca$^{2+}$ overload, resulting in arrhythmia, tachycardia [25], cardiomyopathy and cardiac fibrosis [26]. We found that compared to TMEM43, TMEM43 S358L mutant had lower binding level to both VDAC1 and VDAC2. It has been reported [27] that Tmem43 interacts with VDACs at the endoplasmic/sarcoplasmic reticulum-mitochondrial (ER/SR-mitochondrial) contact sites, and this interaction is essential for normal mitochondrial function. The decreasing binding of VDACs in TMEM43 mutant H9c2 cells was further validated by IF and IP. More importantly, TMEM43 mutant led to mitochondrial fragmentation and mitochondrial membrane potential loss. Transient opening of mitochondrial permeability transition pore (mPTP) contributes rapid Ca$^{2+}$ efflux to protect cells from oxidative damage under physiological conditions, while sustained mPTP opening causes mitochondrial damage, cell death and is involved in a range of pathologies [28,29]. The pore opening is under the regulation of reactive oxygen species, Ca$^{2+}$ ions, and changes of mitochondrial membrane potential [30]. The aberrant opening of mPTP can cause ischemia–reperfusion injury of the heart, which makes the mPTP a potential target for therapeutic intervention [31,32]. Not only in heart diseases but also for neurodegeneration, inhibition of mPTP has become attractive [33]. Decreasing interaction of VDAC2 and TMEM43 in mutant cells may change the cellular localization of VDAC2 [34] and influence mitochondrial Ca$^{2+}$ uptake, leading to effects like arrhythmia, fibrosis that are found in TMEM43 S358L associated ARVC. These studies indicate that TMEM43 likely change heart contraction function through its interaction with critical elements in calcium signaling pathway, especially VDACs.

In the present study, we systematically investigate the protein networks interaction with wild-type TMEM43 and its S358L mutant. Notably, this study reveals that VDAC mediated mitochondrial dysfunctions are relative to ARVC pathology.

## Supporting information

**S1 Fig. IP-MS determined the interaction protein network of TMEM43 and its S358L mutant.** (A) A549 cells stably expressing vector control, HA-Flag-TMEM43 and its S358L mutant were lysed and purified via Flag-agarose beads in 1st run. The elution product of Flag peptides were further purified via HA- agarose beads in 2nd run. The products from each step were subjected to immunoblotting with HA antibody to ensure that the process works well. FT, flow-through; W1, 1st time wash; W2, 2nd time wash; E1, 1st time elution; E2, 2nd time elution; E3, 3rd time elution; Flag, Flag beads; HA, HA beads. (B) SDS-polyacrylamide gel electrophoresis and silver staining of immunoprecipitation (IP) products before mass spectral (MS) analysis. The bands of target protein were only observed on Tmem43 (T43) and Mutant (Mu) samples but not on Vector (Vc) sample. (TIF)

**S2 Fig. Functional pathway enrichment of differentially binding proteins.** (A) The top 20 pathways in KEGG pathway annotation of all DBPs among all the three different groups of samples. The total numbers of identified proteins in every pathway were shown. (B) Scatterplot of enriched gene ontology (GO) pathways of DBPs in TMEM43 versus TMEM43pS358L mutant. (TIF)

**S3 Fig. The interaction of TMEM43 and its S358L mutant with several selected binding partners were confirmed in H9c2 cells via the Immunofluorescence (IF) staining.** H9c2 cells stably expressing HA-Flag-TMEM43 and its S358L mutant were stained with DAPI, HA antibody, and the indicated antibodies including γActin (ACTG1), Connexin 43 (CX43), Caveolin 1 (CAV1) and Emerin (EMD). scale bar: 50um. (TIF)

**S4 Fig. TurboID system validated the interaction of TMEM43 and its mutant with several indicated proteins in H9c2 cells.** (A) The 3XHA-Turbo-TMEM43 and its S358L mutation plasmids (top panel) were transfected into H9c2 cells. Immunoprecipitation (IP) of the cell lysates with streptavidin revealed the pulldown of TMEM43 and its mutant (indicated by HA), and displayed the interaction of Lamin A/C (LMNA), Emerin (EMD) and ATP2A2 with TMEM43 and its mutant. (B) The TurboID Immunofluorescence (IF) staining validated the colocalization of Flag-TMEM43 and its S358L mutant (Biotin, green) with APOB in cardiac myoblast H9c2 cells. H9c2 cells transfected with 3XHA-Turbo-TMEM43 and its S358L mutant were stained with DAPI, streptavidin (biotin staining), and APOB antibodies. The pictures were taken under fluorescent microscope. scale bar: 50um.
(TIF)

**S5 Fig. TurboID system confirmed that TMEM43 and its mutant both partially co-localized with their binding partners in H9c2 cardiac myoblasts.** The TurboID Immunofluorescence (IF) staining validated the colocalization of Flag-TMEM43 and its S358L mutant (green) with the selected binding partners (red) including Sarcoplasmic/Endoplasmic Reticulum Calcium ATPase 2 (SERCA2 or ATP2A2), Emerin (EMD), Lamin A/C (LMNA) and HA in cardiac myoblast H9c2 cells. H9c2 cells transfected with 3XHA-Turbo-TMEM43 and its S358L mutant were stained with DAPI, streptavidin (biotin staining), and the indicated antibodies. The pictures were taken under fluorescent microscope. scale bar: 50um The colocalization percentage in 6 random regions of IF staining under 200 magnifications was counted and compared in bottom panel. Student t-test was performed. ** indicates $p < 0.01$, **** indicates $p < 0.001$.
(TIF)

**S6 Fig. The specific domains of VDACs binding to TMEM43 were characterized.** (A) Vector, Flag-VDAC1/2 full-length (FL) or its specific domains like Aa1–100 were co-transfected with HA -TMEM43 into 293T cells. Immunoprecipitation (IP) of the cell lysates with HA antibody displayed the interaction of VDACs and VDAC1- Aa50–200 with TMEM43. (B) The protein levels of VDAC1, VDAC2 and Histone H3 in the nucleus of H9c2 cells are displayed. (C) The protein levels of TMEM43 and Actin in A549 wild type (WT), A549 TMEM43 (T43) overexpression cells, H9c2 wild type (WT), and H9c2 TMEM43 (T43) overexpression cells.
(TIF)

**S7 Fig. The single channel images of Immunofluorescence (IF) staining of Figure 4D.** DIPA blue, TMEM43 green, VDACs red.
(TIF)

**S8 Fig. TMEM43 interacts with VDACs.** (A) Immunoblotting with indicated antibodies displayed the protein expression of VDAC2, VDAC1 and TMEM43 with or without IP of VDACs. (B) The lysate of mouse hearts was IP with indicated antibodies and submitted to immunoblotting. (C) Immunoblotting with indicated antibodies displayed the protein expression in the nucleus of H9c2 cells with or without TMEM43 mutant.
(TIF)

**S9 Fig. The individual channel of Figure 4A.** TMEM43 is stained with green color. VDACs are stained with red color. The nucleus is stained with blue DAPI.
(TIF)

**S10 Fig. JC-1 staining was applied to detect the mitochondrial membrane potential.** Representative imagesof JC-1 aggregate (red) and JC-1 monomer (green) were presented. Scale bar: 20um.
(TIF)

**S11 Fig. The mitochondrial network of VDAC1&2 immunofluorescence.**
(TIF)

**S1 Table. The differential expression proteins between TMEM43 vs Vector cells identified via IP-MS.**
(XLSX)

**S2 Table. The differential expression proteins between TMEM43 vs Mutant cells identified via IP-MS.**
(XLSX)

**S1 File. Raw images.** The original images of all immunobloting.
(PDF)

## Acknowledgement

We'd like to acknowledge Suzhou Bionovogene (http://www.bionovogene.com) for providing MS analysis.

## Author contributions

**Conceptualization:** Chengming Zhu.

**Data curation:** Qingqing Zhu, Yingsi Lu, Yan Li.

**Formal analysis:** Yizhou Jiang.

**Funding acquisition:** Guoxing Zheng, Yi Lu, Jian Zhang, Chengming Zhu.

**Investigation:** Yingsi Lu, Yizhou Jiang, Yan Li, Hong Chen, Lifen Huang, Nannan Tang.

**Methodology:** Qingqing Zhu, Yingsi Lu.

**Project administration:** Guoxing Zheng.

**Resources:** Yi Lu, Jian Zhang, Chengming Zhu.

**Supervision:** Guoxing Zheng, Chengming Zhu.

**Validation:** Qingqing Zhu, Yingsi Lu, Hong Chen.

**Visualization:** Qingqing Zhu.

**Writing – original draft:** Guoxing Zheng.

**Writing – review & editing:** Guoxing Zheng, Bo Li.

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
