## [Decision Letter · Decision Letter 0]

10 Mar 2025

Dear Dr. Zheng,

Thank you for submitting your manuscript to PLOS ONE. After careful consideration, we feel that it has potential merit but does not fully meet PLOS ONE’s publication criteria as it currently stands. Therefore, we invite you to submit a revised version of the manuscript that addresses the points raised during the review process.

We look forward to receiving your revised manuscript.

Kind regards,

Henry Sutanto, MD, MSc, PhD

Academic Editor

PLOS ONE

“This work was supported by grants from Shenzhen Science and Technology Planning Program (JCYJ20210324122814040 to G. Zheng, JCYJ20190809161811237 to J. Zhang and JCYJ20210324104214040 to Yi Lu), National Natural Science Foundation of China (81972420 to Yi Lu and 81972766, 82173336 to J. Zhang) and Chengming Zhu’s start-up funding from Sun Yat-Sen University.”

In your cover letter, please note whether your blot/gel image data are in Supporting Information or posted at a public data repository, provide the repository URL if relevant, and provide specific details as to which raw blot/gel images, if any, are not available. Email us at plosone@plos.org if you have any questions

Additional Editor Comments:

While the authors have provided clear and detailed methodologies, there are major concerns raised by the Reviewers that must be addressed. I concur that additional, extensive, and repeated experiments are necessary to strengthen the validity of the findings. To substantiate their claims, the authors must not only address the Reviewers' comments but also provide conclusive evidence that the TMEM43 S358L mutation causes ARVC5. This requires in vivo experiments rather than relying solely on in vitro and ex vivo approaches.

In detail, the authors must provide direct evidence linking the TMEM43 S358L mutation to ARVC. This could be achieved through in vivo studies using a transgenic mouse model that demonstrates this mutation induces the structural, functional, and electrophysiological remodeling characteristic of ARVC5. At the very least, in vitro electrophysiological characterization (e.g., patch-clamp techniques) should be performed to support their claims. At this stage, the assertion that 'characterizing the interacting proteins of TMEM43 mutants is important for unraveling the underlying mechanisms of ARVC5' remains unsubstantiated. While the study explores the relationship between the S358L mutation and VDAC1/VDAC2, this connection alone does not establish causality or provide sufficient insight into whether this mutation is indeed responsible for ARVC5.

Reviewers' comments:

Reviewer's Responses to Questions

**Comments to the Author**

1. Is the manuscript technically sound, and do the data support the conclusions?

Reviewer #1: Partly

Reviewer #2: Yes

Reviewer #3: Yes

2. Has the statistical analysis been performed appropriately and rigorously?

Reviewer #1: No

Reviewer #2: Yes

Reviewer #3: Yes

3. Have the authors made all data underlying the findings in their manuscript fully available?

Reviewer #1: Yes

Reviewer #2: Yes

Reviewer #3: Yes

4. Is the manuscript presented in an intelligible fashion and written in standard English?

Reviewer #1: Yes

Reviewer #2: Yes

Reviewer #3: Yes

Reviewer #1: The authors performed IP-MS between TMEM43 and TMEM43 S358L transfected cells and found a different protein binding profile with TMEM43. Among all the interacted proteins, they identified proteins related to calcium signaling, lipid metabolism, and cardiovascular diseases. Following in vivo verification, they discovered that VDAC may play a key role in pathogenesis. The mutant TMEM43 group exhibited a decreased binding interaction between VDAC and TMEM. The paper is concise, but significant revisions need to be addressed for further review.

Please see below for details.

Major:

For IP-MS, IgG control is needed for immunoprecipitation (IP) mass spectrometry to help identify false-positive interactions.

A few quantifications are lacking (Fig S4B, S5, 4A). It is recommended that every co-localization IF image should followed by quantification analysis. It is important to point out that the analysis is done in individually successfully plasmid-transfected cells. Due to the existence of endogenous TMEM43, it is difficult to separate the phenotype derived from TMEM43 mutant or endogenously. Making an endogenous model on TMEM43 mutant is highly recommended.

DAPI is often not mentioned in the figure legend where IF was described.

Figure 4:

1. VDAC is a transmembrane calcium channel protein expressed in the mitochondrial outer membrane. In Figure 4A, the red fluorescence of VDAC2 is from the nucleus. VDAC1 looks normal.

2. Why is HA localization very different from Figure 3? Isn’t TMEM localized in the nuclear membrane? In Figure 4A, HA is also expressed in the cytosol.

Therefore, the authors can’t make convincing results from Figure 4A.

Figure 4B, IgG control is needed.

Figure 4C, the description is not clear. Labeling what the samples are and what antibodies are used to precipitate is recommended.

For input samples, the TMEM43 density is too low to tell. Either increase the loading amount or use a more sensitive ECL.

All Western images need to be labeled the molecular size ladder; please refer to Western image publication standards. In the supplementary, please upload all raw images with appropriate labeling.

To conclude, the decreased interaction between VDAC and TMEM needs to be quantified. Another method is highly recommended to make the conclusion convincing. If authors can make IF results convincing, it also counts. Stain cells with mitochondria maker to see whether mitochondria co-localization with the nucleus decrease may be helpful as VDAC is expressed in mitochondria.

Figure 4A, I recommend having an individual channel in addition to a merged one; for example, the red fluorescence in the left bottom is too dim to tell whether there is some localization on the nuclear membrane or TMEM43.

There are limitations of IF and IP. IF is a 2D image, and proteins are inseparable between the cytosol and organelles if a generic protein lysis method is used. To make the data more convincing, nuclear protein-specific isolation is recommended to test whether the mutant group's intersection between VDAC and TMEM43 is decreased.

Figure 5:

A did not label what violet signal is.

Line 467, it is abrupt to mention TMEM43 function in mitochondria, it is not clearly described in the context.

B and C

Analysis should be done in the cells that have the plasmid transfected. Not all cells have the TMEM mutant plasmid.

Figure B needs a higher resolution to indicate that JC-1 is in the mitochondria to prove its efficacy.

Minor:

It is vague to say:

“IF staining of TurboID H9c2 cells revealed that biotin and HA labeled TMEM43 were completely overlapping (Fig. S5), indicating the precision of TurboID system.”

Are authors implying Turbo ID system consistency with Flag-tagged TMEM43 used before, since plasmids in both systems are HA-tagged?

The expression plasmid information in the flag tagged TMEM43 has not been revealed.

Reviewer #2: The authors use IP-MS technique and identify 166 interacting protein candidates of TMEM43 mutants. They further confirm that TMEM43 mutant reduce its interaction with VDAC, which mediates mitochondrial dysfunction in cardiac myoblast H9c2 cells of TMEM43 pS358L. This work unravels the underlying mechanisms of TMEM43 mutant in ARVC5, and implies a critical role of mitochondria in ARVC pathology.

However, there are several minor concerns should be considered:

-VDAC is a mitochondrial outer membrane protein, the authors should explain the possibility how VDAC interacts with TMEM43.

-In Figure 4B, the input of VDAC2 is missing.

-In Figure 4C, the input of VDAC1 in TMEM43 group and all the TMEM43 bands are weak. The authors should explain this or repeat this experiment.

-The IP of TMEM43 with VDAC2 should also be added.

-The mitochondrial matrix Ca2+ level should be detected.

-In Discussion, “TMEM43 mutant led to 515 mitochondrial fragmentation and mitochondrial membrane potential loss”. The role of mitochondrial permeability transition pore should be discussed.

Reviewer #3: The Present manuscript authors found that mutant TMEM43 shows reduced interaction with the VDAC which eventually causes the mitochondria dysfunction. The work was conducted thoroughly, and the manuscript was crafted well. However, there are some key concerns before considering it for publication.

1. The manuscript title states, “Proteomic screening of TMEM43 binding partners identifies VDAC leading to mitochondrial dysfunction in Arrhythmic Right Ventricular Cardiomyopathy.” However, the study primarily focuses on the interaction differences between WT and mutant TMEM43, which is essentially a protein characterization study. Since the research was not conducted using an ARVC mouse model or patient primary cells, I suggest that the authors revise the title accordingly.

2. Please correct the mistake of cell number in line number 250.

3. Figure 3: It is recommended to include a control image (immunofluorescence performed in cells without the HA tag) and provide high-quality images to enhance the overall quality of the manuscript.

4. Figure 4A: The VDAC1&2 immunofluorescence shows mitochondrial puncta formation, indicating that the cells were not in a healthy state. I recommend that the authors include images where VDAC1&2 immunofluorescence demonstrates the mitochondrial network.

5. Figures 4B & 4C: I strongly recommend that the authors perform a reverse IP using VDAC1&2 in H9c2 cells and mouse heart lysate.

6. Figure 5A: It would be beneficial to include high-quality images where mitochondrial morphology is clearly visible, with a greater number of cells.

7. Figure 5: It is recommended to measure ROS levels in both the mutant and wild-type, as it is a key hallmark of mitochondrial dysfunction.

8. In line number 471, JC-1 staining is incorrectly referred to as immunofluorescence. Please correct the sentence accordingly.

**Do you want your identity to be public for this peer review?** For information about this choice, including consent withdrawal, please see our Privacy Policy

Reviewer #1: **Yes: ** Zhuqing Liang

Reviewer #2: No

Reviewer #3: **Yes: ** Arun Kumar Paripati

---

## [Author Response · Author response to Decision Letter 1]

30 Jul 2025

Response to the comments

Summary of revisions

1.Several new experiments were carried out to address the comments including detection of Ca2+ level and ROS level in S10 Fig, S8 Fig Reverse IP with VDACs in H9c2 cells and mouse hearts, IgG IP in Fig 4C, WB in S6 Fig et al.

2.Six panels including Fig 3A, Fig 4A, Fig 4C, Fig 4D, Fig 5A, Fig 5B are all updated.

3.Quantity analyses are added in Fig 4B and S5 Fig.

4.New result are provided inS6 Fig, S8 Fig, S9 Fig, S10 Fig and S11 Fig.

5.The marker and sample names are labeled in all original gel images in S1 File.

6.The whole manuscript is revised thoroughly to meet the style requirements of PLOS ONE.

Response: The manuscript is revised to meet the style requirements of PLOS ONE.

“This work was supported by grants from Shenzhen Science and Technology Planning Program (JCYJ20210324122814040 to G. Zheng, JCYJ20190809161811237 to J. Zhang and JCYJ20210324104214040 to Yi Lu), National Natural Science Foundation of China (81972420 to Yi Lu and 81972766, 82173336 to J. Zhang) and Chengming Zhu’s start-up funding from Sun Yat-Sen University.”

Response: The funders had no role, and the statement is added in the manuscript.

Response: The ethics statement is moved to the Methods section.

In your cover letter, please note whether your blot/gel image data are in Supporting Information or posted at a public data repository, provide the repository URL if relevant, and provide specific details as to which raw blot/gel images, if any, are not available. Email us at plosone@plos.org if you have any questions

Response: The original uncropped and unadjusted images are provided in the Supporting Information S1 File, please check it.

Additional Editor Comments:

While the authors have provided clear and detailed methodologies, there are major concerns raised by the Reviewers that must be addressed. I concur that additional, extensive, and repeated experiments are necessary to strengthen the validity of the findings. To substantiate their claims, the authors must not only address the Reviewers' comments but also provide conclusive evidence that the TMEM43 S358L mutation causes ARVC5. This requires in vivo experiments rather than relying solely on in vitro and ex vivo approaches.

In detail, the authors must provide direct evidence linking the TMEM43 S358L mutation to ARVC. This could be achieved through in vivo studies using a transgenic mouse model that demonstrates this mutation induces the structural, functional, and electrophysiological remodeling characteristic of ARVC5. At the very least, in vitro electrophysiological characterization (e.g., patch-clamp techniques) should be performed to support their claims. At this stage, the assertion that 'characterizing the interacting proteins of TMEM43 mutants is important for unraveling the underlying mechanisms of ARVC5' remains unsubstantiated. While the study explores the relationship between the S358L mutation and VDAC1/VDAC2, this connection alone does not establish causality or provide sufficient insight into whether this mutation is indeed responsible for ARVC5.

Response: The direct evidence in rodent models linking the TMEM43 S358L mutation to ARVC has been published by 3 different groups (Protein&Cell 2019 knock-in mice [1]; Circulation 2019 transgenic mice [2]; American Journal of Physiology Heart and Circulatory Physiology 2023 knock-in mice [3]). The structural abnormalities of TMEM43 S358L mice in all 3 publications contained significantly higher ratio of heart weight to body weight (HW/BW), progressive cardiac dilatation, and faster fibrofatty progression. The echocardiography analyses of cardiac functions showed reduced left ventricular ejection fraction (LVEF) in mutant mice. The electrocardiograms of mutant mice [2, 3] exhibited prolonged QRS interval, and other abnormalities like decreasing QRS amplitude, premature atrial and ventricular contractions, and intolerance strenuous running stress. Moreover, human induced pluripotent stem cell (hiPSC) derived cardiomyocytes bearing the p.S358L mutation [2] showed increased contraction duration, decreased contraction amplitude, and slower Ca2+ transient upon isoproterenol treatments. It is interesting that TMEM43 interacted with VDAC1 in other cell lines [4]. Furthermore, the mitochondria abnormalities were discovered in the mouse hearts carrying TMEM43 S358L mutation [3]. Whatever, the assertion is revised.

Reference

[1]Zheng G, Jiang C, Li Y, Yang D, Ma Y, Zhang B, Li X, Zhang P, Hu X, Zhao X, Du J, Lin X. TMEM43-S358L mutation enhances NF-κB-TGFβ signal cascade in arrhythmogenic right ventricular dysplasia/cardiomyopathy. Protein Cell. 2019 Feb;10(2):104-119. doi: 10.1007/s13238-018-0563-2. Epub 2018 Jul 6. PMID: 29980933; PMCID: PMC6340891.

[2]Padrón-Barthe L, Villalba-Orero M, Gómez-Salinero JM, Domínguez F, Román M, Larrasa-Alonso J, Ortiz-Sánchez P, Martínez F, López-Olañeta M, Bonzón-Kulichenko E, Vázquez J, Martí-Gómez C, Santiago DJ, Prados B, Giovinazzo G, Gómez-Gaviro MV, Priori S, Garcia-Pavia P, Lara-Pezzi E. Severe Cardiac Dysfunction and Death Caused by Arrhythmogenic Right Ventricular Cardiomyopathy Type 5 Are Improved by Inhibition of Glycogen Synthase Kinase-3β. Circulation. 2019 Oct;140(14):1188-1204. doi: 10.1161/CIRCULATIONAHA.119.040366. Epub 2019 Sep 5. PMID: 31567019; PMCID: PMC6784777.

[3]Orgil BO, Munkhsaikhan U, Pierre JF, Li N, Xu F, Alberson NR, Johnson JN, Wetzel GT, Boukens BJD, Lu L, Towbin JA, Purevjav E. The TMEM43 S358L mutation affects cardiac, small intestine, and metabolic homeostasis in a knock-in mouse model. Am J Physiol Heart Circ Physiol. 2023 Jun 1;324(6):H866-H880. doi: 10.1152/ajpheart.00712.2022. Epub 2023 Apr 21. PMID: 37083466; PMCID: PMC10190833.

[4]Zhang N, Wang F, Yang X, Wang Q, Chang R, Zhu L, Feitelson MA, Chen Z. TMEM43 promotes the development of hepatocellular carcinoma by activating VDAC1 through USP7 deubiquitination. Transl Gastroenterol Hepatol. 2024 Jan 25;9:9. doi: 10.21037/tgh-23-108. PMID: 38317750; PMCID: PMC10838614.

5. Review Comments to the Author

Reviewer #1: The authors performed IP-MS between TMEM43 and TMEM43 S358L transfected cells and found a different protein binding profile with TMEM43. Among all the interacted proteins, they identified proteins related to calcium signaling, lipid metabolism, and cardiovascular diseases. Following in vivo verification, they discovered that VDAC may play a key role in pathogenesis. The mutant TMEM43 group exhibited a decreased binding interaction between VDAC and TMEM. The paper is concise, but significant revisions need to be addressed for further review.

Please see below for details.

Major:

For IP-MS, IgG control is needed for immunoprecipitation (IP) mass spectrometry to help identify false-positive interactions.

Response: Although IgG control is not included, the empty vector controls are contained in IP-MS assay (S1 Table, S2 Table, Fig 1B, S1 Fig). This vector controls help to exclude the false-positive interactions.

A few quantifications are lacking (Fig S4B, S5, 4A). It is recommended that every co-localization IF image should followed by quantification analysis. It is important to point out that the analysis is done in individually successfully plasmid-transfected cells. Due to the existence of endogenous TMEM43, it is difficult to separate the phenotype derived from TMEM43 mutant or endogenously. Making an endogenous model on TMEM43 mutant is highly recommended.

DAPI is often not mentioned in the figure legend where IF was described.

Response: The co-localization IF images are all followed by quantification analyses. DAPI is mentioned in all figure legends of immunofluorescence. We compared the endogenous TMEM43 protein level with overexpression ones and found that the endogenous expression is much lower than the overexpression (S6C Fig). Thus, the effect of endogenous TMEM43 is quite limited due to the very low expression.

Figure 4:

1.VDAC is a transmembrane calcium channel protein expressed in the mitochondrial outer membrane. In Figure 4A, the red fluorescence of VDAC2 is from the nucleus. VDAC1 looks normal.

Response: We checked the cellular location of VDAC2 in Gencards (https://www.genecards.org/) and found that VDAC2 also locates in the nucleus as shown in the following figure.

2. Why is HA localization very different from Figure 3? Isn’t TMEM localized in the nuclear membrane? In Figure 4A, HA is also expressed in the cytosol.

Therefore, the authors can’t make convincing results from Figure 4A.

Response: Although it is not so obvious, HA is also expressed in cytosol in Figure3. The different staining intensity of HA may be due to the difference of cell lines of A549 and H9c2. As indicated by the cellular location of TMEM43 in Gencards (https://www.genecards.org/), the cytosol staining of HA may come from endoplasmic reticulum that surround the nucleus.

Figure 4B, IgG control is needed.

Response: The IgG control is added in Fig 4C (original 4B).

Figure 4C, the description is not clear. Labeling what the samples are and what antibodies are used to precipitate is recommended.

For input samples, the TMEM43 density is too low to tell. Either increase the loading amount or use a more sensitive ECL.

Response: Antibodies used to precipitate are labeled. The input of TMEM43 is updated in this figure.

All Western images need to be labeled the molecular size ladder; please refer to Western image publication standards. In the supplementary, please upload all raw images with appropriate labeling.

Response: All original Western images labeled with the molecular size ladder are provided in the supplementary material named S1 File.

To conclude, the decreased interaction between VDAC and TMEM needs to be quantified. Another method is highly recommended to make the conclusion convincing. If authors can make IF results convincing, it also counts. Stain cells with mitochondria maker to see whether mitochondria co-localization with the nucleus decrease may be helpful as VDAC is expressed in mitochondria.

Response: The reverse IP with VDACs in H9c2 and mouse hearts were performed as shown in S8 Fig. Since VDAC is highly expressed in both nucleus and mitochondria, detection of mitochondria co-localization with the nucleus may not help.

Figure 4A, I recommend having an individual channel in addition to a merged one; for example, the red fluorescence in the left bottom is too dim to tell whether there is some localization on the nuclear membrane or TMEM43.

Response: The individual channel of Figure 4A are provided in S9 Fig in the file of supplementary materials.

There are limitations of IF and IP. IF is a 2D image, and proteins are inseparable between the cytosol and organelles if a generic protein lysis method is used. To make the data more convincing, nuclear protein-specific isolation is recommended to test whether the mutant group's intersection between VDAC and TMEM43 is decreased.

Response: We tried to isolate the nucleus in H9c2 cells and detected the protein level of VDACs. However, the signal is quite weak through Western blot (S6B Fig). Thus, there is technical problem to compare the nuclear protein level of VDACs in T43 and Mu cells.

Figure 5:

A did not label what violet signal is.

Line 467, it is abrupt to mention TMEM43 function in mitochondria, it is not clearly described in the context.

Response: Blue is DAPI staining, which is mentioned in figure legend. The description in the context is modified.

B and C

Analysis should be done in the cells that have the plasmid transfected. Not all cells have the TMEM mutant plasmid.

Response: We are confused, please make it clear.

Figure B needs a higher resolution to indicate that JC-1 is in the mitochondria to prove its efficacy.

Response: Figure 5B is updated.

Minor:

It is vague to say:

“IF staining of TurboID H9c2 cells revealed that biotin and HA labeled TMEM43 were completely overlapping (Fig. S5), indicating the precision of TurboID system.”

Are authors implying Turbo ID system consistency with Flag-tagged TMEM43 used before, since plasmids in both systems are HA-tagged?

The expression plasmid information in the flag tagged TMEM43 has not been revealed.

Response: It means that TurboID system is consistent to IF staining. The description is revised. The plasmid information of Flag tagged TMEM43 is provided in the section of material and methods, that is the lentivirus p2k7 vector carrying HA-Flag-TMEM43 or HA-Flag-TMEM43 S358L mutant.

Reviewer #2: The authors use IP-MS technique and identify 166 interacting protein candidates of TMEM43 mutants. They further confirm that TMEM43 mutant reduce its interaction with VDAC, which mediates mitochondrial dysfunction in cardiac myoblast H9c2 cells of TMEM43 pS358L. This work unravels the underlying mechanisms of TMEM43 mutant in ARVC5, and implies a critical role of mitochondria in ARVC pathology.

However, there are several minor concerns should be considered:

-VDAC is a mitochondrial outer membrane protein, the authors should explain the possibility how VDAC interacts with TMEM43.

Response: We checked the cellular locations of VDAC in Gencards (https://www.genecards.org/) and found that the expression of VDACs are highest both in the mitochondria and nucleus with confidence 5. This indicates that VDAC may interacts with TMEM43 in the nucleus.

-In Figure 4B, the input of VDAC2 is missing.

Response: VDAC2 input is added.

-In Figure 4C, the input of VDAC1 in TMEM43 group and all the TMEM43 bands are weak. The authors should explain this or repeat this experiment.

Response: The figure is updated.

-The IP of TMEM43 with VDAC2 should also be added.

Response: The result is shown in S8 Fig. The endogenous IP of VDACs in H9c2 and mouse hearts work not very well.

-The mitochondrial matrix Ca2+ level should be detected.

Response: The mitochondrial matrix Ca2+ level was detected through Fluo-4 AM fluorescent probe. The results are s

---

## [Decision Letter · Decision Letter 1]

7 Aug 2025

Dear Dr. Zheng,

Thank you for submitting your manuscript to PLOS ONE. After careful consideration, we feel that it has merit but does not fully meet PLOS ONE’s publication criteria as it currently stands. Therefore, we invite you to submit a revised version of the manuscript that addresses the points raised during the review process.

**Thank you for your resubmission and detailed response.**

**We have completed our reevaluation of your manuscript and appreciate the improvements made. However, several important issues remain unresolved. Should you be able to address these concerns satisfactorily, we would be happy to reconsider your submission.**

**Additionally, please ensure that you include a version of the manuscript with tracked changes from the original submission (R0). It appears that the revisions in the current version were not properly tracked, which hinders our ability to assess the changes accurately.**

**We look forward to receiving your revised submission**

We look forward to receiving your revised manuscript.

Kind regards,

Henry Sutanto, MD, MSc, PhD

Academic Editor

PLOS ONE

Journal Requirements:

Additional Editor Comments:

Thank you for your resubmission and detailed response.

We have completed our reevaluation of your manuscript and appreciate the improvements made. However, several important issues remain unresolved. Should you be able to address these concerns satisfactorily, we would be happy to reconsider your submission.

Additionally, please ensure that you include a version of the manuscript with tracked changes from the original submission (R0). It appears that the revisions in the current version were not properly tracked, which hinders our ability to assess the changes accurately.

We look forward to receiving your revised submission.

Reviewers' comments:

Reviewer's Responses to Questions

**Comments to the Author**

Reviewer #2: (No Response)

Reviewer #3: All comments have been addressed

2. Is the manuscript technically sound, and do the data support the conclusions?

Reviewer #2: (No Response)

Reviewer #3: Yes

3. Has the statistical analysis been performed appropriately and rigorously?

Reviewer #2: (No Response)

Reviewer #3: Yes

4. Have the authors made all data underlying the findings in their manuscript fully available?

Reviewer #2: (No Response)

Reviewer #3: Yes

5. Is the manuscript presented in an intelligible fashion and written in standard English?

Reviewer #2: (No Response)

Reviewer #3: Yes

Reviewer #2: There are still some concerns to be addressed:

- VDAC is a mitochondrial outer membrane protein. The search in genecard is not sufficient to prove that VDAC can both localize to mitochondria and nucleus. The authors should search in literature and cite the related findings. Moreover, only based on this, the authors cannot conclude that “indicates that VDAC may interacts with TMEM43 in the nucleus”, because some nuclear proteins are able to translocate to mitochondria. In addition, the authors should perform immunoblotting analysis of VDAC in subcellular fraction.

-If the authors conclude VDAC interacts with TMEM43 in the nucleus, what is the molecular mechanism through which TMEM43 regulates mitochondrial dysfunction?

-In the original Figure 4C, the input of VDAC1 in TMEM43 group and all the TMEM43 bands are weak. The authors only changed the blots of input but not the corresponding blots of IP, which is not convincing. The authors should replace all the blots in the original Figure 4C with new experimental data.

-The authors should describe the mitochondrial Ca2+ level measurement and explain why there is no difference between WT and TMEM43 mutant.

-The role of mitochondrial permeability transition pore in the discussion is not sufficient.

Reviewer #3: (No Response)

**Do you want your identity to be public for this peer review?** For information about this choice, including consent withdrawal, please see our Privacy Policy

Reviewer #2: No

Reviewer #3: **Yes: ** Arun Kumar Paripati

---

## [Author Response · Author response to Decision Letter 2]

14 Oct 2025

Reviewer #2: There are still some concerns to be addressed:

- VDAC is a mitochondrial outer membrane protein. The search in genecard is not sufficient to prove that VDAC can both localize to mitochondria and nucleus. The authors should search in literature and cite the related findings. Moreover, only based on this, the authors cannot conclude that “indicates that VDAC may interacts with TMEM43 in the nucleus”, because some nuclear proteins are able to translocate to mitochondria.

Response Sure, VDAC1 is a mitochondrial membrane protein that interact with other partners in mitochondria [1]. Hanna Galganska et al. predicted the communication of VDAC between mitochondria and nucleus [2]. Moreover, Sara de Mateo et al. discovered that VDAC1 and VDAC2 both located in human sperm nucleus through LC-MS/MS [3]. Our immunoblotting analysis showed expression of VDACs proteins in the nucleus (S8C Fig). TMEM43 locates to nucleus inner membrane [4]. Another possibility is that TMEM43 translocates to mitochondria and interacts with VDAC there. Therefore, we do not claim their interaction in the nucleus in the manuscript.

Reference

[1]Shanmughapriya S, Rajan S, Hoffman NE, Higgins AM, Tomar D, Nemani N, Hines KJ, Smith DJ, Eguchi A, Vallem S, Shaikh F, Cheung M, Leonard NJ, Stolakis RS, Wolfers MP, Ibetti J, Chuprun JK, Jog NR, Houser SR, Koch WJ, Elrod JW, Madesh M. SPG7 Is an Essential and Conserved Component of the Mitochondrial Permeability Transition Pore. Mol Cell. 2015 Oct 1;60(1):47-62. doi: 10.1016/j.molcel.2015.08.009. Epub 2015 Sep 17. PMID: 26387735; PMCID: PMC4592475.

[2]Galganska H, Karachitos A, Wojtkowska M, Stobienia O, Budzinska M, Kmita H. Communication between mitochondria and nucleus: putative role for VDAC in reduction/oxidation mechanism. Biochim Biophys Acta. 2010 Jun-Jul;1797(6-7):1276-80. doi: 10.1016/j.bbabio.2010.02.004. Epub 2010 Feb 6. PMID: 20144586.

[3]de Mateo S, Castillo J, Estanyol JM, Ballescà JL, Oliva R. Proteomic characterization of the human sperm nucleus. Proteomics. 2011 Jul;11(13):2714-26. doi: 10.1002/pmic.201000799. Epub 2011 Jun 1. PMID: 21630459.

[4]Bengtsson L, Otto H. LUMA interacts with emerin and influences its distribution at the inner nuclear membrane. J Cell Sci. 2008 Feb 15;121(Pt 4):536-48. doi: 10.1242/jcs.019281. Epub 2008 Jan 29. PMID: 18230648.

In addition, the authors should perform immunoblotting analysis of VDAC in subcellular fraction.

Response: The immunoblotting analysis of VDACs in the nucleus (S8C Fig) showed that both VDACs expressed in the nucleus of H9c2 cells. And the IF staining of VDAC is overlapping with DAPI (Fig 4A).

-If the authors conclude VDAC interacts with TMEM43 in the nucleus, what is the molecular mechanism through which TMEM43 regulates mitochondrial dysfunction?

Response: The statement of “VDAC interacts with TMEM43 in the nucleus” is not included in the manuscript. Here is the hypothesis only for discussion with the reviewer, which is not even mentioned in the manuscript. Reduced binding of VDACs with TMEM43 mutant in the nucleus leads to higher level of VDACs in mitochondria, triggering excessive mPTP opening. Finally, this results in lower mitochondrial matrix Ca2+ level, higher mitochondrial ROS level and lost of mitochondrial membrane potential in mutation cells. Another possibility is not excluded. Wild type TMEM43 translocates from nucleus to mitochondria and interacts with VDACs. Mutation of TMEM43 reduces its binding to VDACs, leads to higher level of VDACs in mitochondria, and triggers excessive mPTP opening. Eventually, it results in the dysfunctions of mitochondria. Similar roles were identified in Hexokinase 1 (HK1) [5]. The decreased binding of VDAC1 to mutant HK1 led to lower level of mitochondrial matrix Ca2+.

Reference

[5]Ceprian M, Juntas-Morales R, Campbell G, Walther-Louvier U, Rivier F, Camu W, Esselin F, Echaniz-Laguna A, Stojkovic T, Bouhour F, Latour P, Tricaud N. The Hexokinase 1 5'-UTR Mutation in Charcot-Marie-Tooth 4G Disease Alters Hexokinase 1 Binding to Voltage-Dependent Anion Channel-1 and Leads to Dysfunctional Mitochondrial Calcium Buffering. Int J Mol Sci. 2024 Apr 15;25(8):4364. doi: 10.3390/ijms25084364. PMID: 38673950; PMCID: PMC11050395.

-In the original Figure 4C, the input of VDAC1 in TMEM43 group and all the TMEM43 bands are weak. The authors only changed the blots of input but not the corresponding blots of IP, which is not convincing. The authors should replace all the blots in the original Figure 4C with new experimental data.

Response: All the blots in the original Figure 4C are replaced with new experimental data in Fig 4D.

-The authors should describe the mitochondrial Ca2+ level measurement and explain why there is no difference between WT and TMEM43 mutant.

Response: The mitochondrial matrix Ca2+ level was detected through Fluo-4 AM fluorescent probe, please refer to “Detection of Ca2+ level” in method section at page15. The results are renewed in Fig 5C. The mitochondrial matrix Ca2+ level decreased in H9c2 cells of TMEM43 mutant.

-The role of mitochondrial permeability transition pore in the discussion is not sufficient.

Response: The roles of mPTP are demonstrated at page 23 with 6 references.

---

## [Decision Letter · Decision Letter 2]

27 Oct 2025

Dear Dr. Zheng,

Thank you for submitting your manuscript to PLOS ONE. After careful consideration, we feel that it has merit but does not fully meet PLOS ONE’s publication criteria as it currently stands. Therefore, we invite you to submit a revised version of the manuscript that addresses the points raised during the review process.

We look forward to receiving your revised manuscript.

Kind regards,

Henry Sutanto, MD, MSc, PhD

Academic Editor

PLOS ONE

Journal Requirements:

Additional Editor Comments (if provided):

Thank you for submitting the revised manuscript. The revision has been carefully reviewed, and the Reviewer has provided a few minor comments that still need to be addressed. The authors are kindly requested to respond to these comments and revise the manuscript accordingly at their earliest convenience, prior to the final decision on the submission.

Reviewers' comments:

Reviewer's Responses to Questions

**Comments to the Author**

Reviewer #2: (No Response)

2. Is the manuscript technically sound, and do the data support the conclusions?

Reviewer #2: (No Response)

3. Has the statistical analysis been performed appropriately and rigorously?

Reviewer #2: (No Response)

4. Have the authors made all data underlying the findings in their manuscript fully available?

Reviewer #2: (No Response)

5. Is the manuscript presented in an intelligible fashion and written in standard English?

Reviewer #2: (No Response)

Reviewer #2: The authors have addressed most of my concerns, but there are still some minor revisions needed:

1) Mitochondria are closely associated with nucleus. The purity of nucleus in S8C Fig need to be checked using mitochondrial markers like TOM20 and ANT etc.

2) “IF staining of VDAC is overlapping with DAPI” is not sufficient to conclude that VDAC can be localized in nucleus but may be just associated or interacted with nucleus.

3) The statement of “VDAC interacts with TMEM43 in the nucleus” is not included in the manuscript. This hypothesis should be discussed in the discussion section.

4) Mutation of TMEM43 reduces its binding to VDACs, leads to higher level of VDACs in mitochondria, and triggers excessive mPTP opening. Is there any evidence that high VDAC can trigger PTP opening?

**Do you want your identity to be public for this peer review?** For information about this choice, including consent withdrawal, please see our Privacy Policy

Reviewer #2: No

---

## [Author Response · Author response to Decision Letter 3]

29 Nov 2025

Reviewer #2: The authors have addressed most of my concerns, but there are still some minor revisions needed:

1)Mitochondria are closely associated with nucleus. The purity of nucleus in S8C Fig need to be checked using mitochondrial markers like TOM20 and ANT etc.

Response:

We thank the reviewer for their valuable comments on this issue. After optimizing and repeating our experiments (≥3 times), we found that it remains technically challenging to completely remove TOM20-positive mitochondrial proteins from nuclear lysates.

Interestingly, a German group recently reported that [1] “Drosophila Tmem43 is localized at the ER/SR (endoplasmic reticulum/sarcoplasmic reticulum) membrane and interacts with the outer mitochondrial membrane protein Porin/VDAC. This interaction is lost in a Tmem43 p.S333L mutant that resembles the human p.S358L mutation. In addition, Tmem43 p.S333L caused a breakdown in mitochondrial membrane potential and increased cellular reactive oxygen species.”

One possible explanation for the presence of TOM20 in wild-type cells but not in mutant cells is that ER/SR–mitochondria complexes are present in the nuclear lysates of wild-type TMEM43 cells, whereas the TMEM43 mutation disrupts ER/SR–mitochondria association, leading to the loss of mitochondria from nuclear lysates.

Based on these findings, we propose the following molecular mechanism: under physiological conditions, TMEM43 located in the ER/SR interacts with VDACs on the mitochondrial membrane. The TMEM43 p.S358L mutation impairs this binding capacity, thereby disrupting ER/SR–mitochondria tethering. This, in turn, triggers mitochondrial dysfunction, manifested as mitochondrial fragmentation, loss of membrane potential, reduced Ca²⁺ levels, and elevated ROS.

2)“IF staining of VDAC is overlapping with DAPI” is not sufficient to conclude that VDAC can be localized in nucleus but may be just associated or interacted with nucleus.

Response: It is possible that VDAC may be just associated or interacted with nucleus.

3)The statement of “VDAC interacts with TMEM43 in the nucleus”is not included in the manuscript. This hypothesis should be discussed in the discussion section.

Response: This hypothesis is updated in the discussion section of updated manuscript. That is “It has been reported that Tmem43 interacts with VDACs at the endoplasmic/sarcoplasmic reticulum-mitochondrial (ER/SR-mitochondrial) contact sites, and this interaction is essential for normal mitochondrial function.”

4)Mutation of TMEM43 reduces its binding to VDACs, leads to higher level of VDACs in mitochondria, and triggers excessive mPTP opening. Is there any evidence that high VDAC can trigger PTP opening?

Response: Indeed, several indirect lines of evidence support a role for elevated VDAC levels in triggering PTP opening. The relationship is evidenced by both gain-of-function and loss-of-function studies:

· On one hand, VDAC1 overexpression induces cell death, and higher VDAC1 levels correlate with increased apoptosis in pathological models [2, 3].

· On the other hand, down-regulating or inhibiting VDAC1 produces protective effects, such as suppressed cell growth and reduced injury during ischemia-reperfusion [2, 4].

· Mechanistically, VDAC inhibition with antibodies prevents pivotal mitochondrial apoptotic events, including cytochrome c release and the collapse of the membrane potential [5].

Based on this evidence, it has been proposed that VDAC is a key contributor to PTP opening, which subsequently leads to mitochondrial damage and cell death [6]. This body of work indirectly supports our observation that increased VDAC levels could facilitate mPTP opening.

References

[1] Jürgens K, Menzel L, Klinke N, Schäper L, Breitsprecher L, Holtmannspötter M, Psathaki OE, Walter S, Ratnavadivel S, Malmendal A, Meyer H, Milting H, Paululat A. The ARVC-5-associated protein TMEM43 controls mitochondrial energy metabolism by stabilising ER-mitochondrial contact sites. Cell Mol Life Sci. 2025 Nov 14;82(1):400. doi: 10.1007/s00018-025-05942-z. PMID: 41236655; PMCID: PMC12618784.

[2] Abu-Hamad, S.; Sivan, S.; Shoshan-Barmatz, V. The expression level of the voltage-dependent anion channel controls life and death of the cell. Proc. Natl. Acad. Sci. USA 2006, 103, 5787–5792.

[3] Sasaki, K.; Donthamsetty, R.; Heldak, M.; Cho, Y.-E.; Scott, B.T.; Makino, A. VDAC: Old protein with new roles in diabetes. Am. J. Physiol. Physiol. 2012, 303, C1055–C1060.

[4] Feng, Y.; Madungwe, N.B.; Imam Aliagan, A.D.; Tombo, N.; Bopassa, J.-C. Liproxstatin-1 protects the mouse myocardium against ischemia/reperfusion injury by decreasing VDAC1 levels and restoring GPX4 levels. Biochem. Biophys. Res. Commun. 2019, 520.

[5] Shimizu S, Matsuoka Y, Shinohara Y, Yoneda Y, Tsujimoto Y. Essential role of voltage-dependent anion channel in various forms of apoptosis in mammalian cells. J Cell Biol. 2001 Jan 22;152(2):237-50. doi: 10.1083/jcb.152.2.237. PMID: 11266442; PMCID: PMC2199613.

[6] Parmar MY, Shah PA, Gao J, Gandhi TR. Hepatoprotection through regulation of voltage dependent anion channel expression by Amomum subulatum Roxb seeds extract. Indian J Pharmacol. 2011 Nov;43(6):671-5. doi: 10.4103/0253-7613.89824. PMID: 22144772; PMCID: PMC3229783.

---

## [Editor Report · Decision Letter 3]

2 Dec 2025

Proteomic screening of TMEM43 binding partners identifies VDAC leading to mitochondrial dysfunction

PONE-D-25-06474R3

Dear Dr. Zheng,

We’re pleased to inform you that your manuscript has been judged scientifically suitable for publication and will be formally accepted for publication once it meets all outstanding technical requirements.

Kind regards,

Henry Sutanto, MD, MSc, PhD

Academic Editor

PLOS ONE

Additional Editor Comments (optional):

Well done! Congratulations.
---

## [Editor Report · Acceptance letter]

PONE-D-25-06474R3

PLOS One

Dear Dr. Zheng,

I'm pleased to inform you that your manuscript has been deemed suitable for publication in PLOS One. Congratulations! Your manuscript is now being handed over to our production team.

Kind regards,

on behalf of

Dr. Henry Sutanto

Academic Editor

PLOS One